# Investigation on Residual Strength and Failure Mechanism of the Ceramic/UHMWPE Armors after Ballistic Tests

**DOI:** 10.3390/ma15030901

**Published:** 2022-01-25

**Authors:** Zhiyong Chen, Yingqiang Xu, Miaoling Li, Bin Li, Weizhi Song, Li Xiao, Yulong Cheng, Songyan Jia

**Affiliations:** 1School of Mechanical Engineering, Northwestern Polytechnical University, Xi’an 710072, China; 1854@163.com (Z.C.); xiaoli_nwpu@163.com (L.X.); 2School of Intelligent Manufacturing, Luoyang Institute of Science and Technology, Luoyang 471023, China; miaolingli1970@163.com (M.L.); libinman@gmail.com (B.L.); wz_song@whut.edu.cn (W.S.); C1679580323@126.com (Y.C.); J1470420348@163.com (S.J.)

**Keywords:** ceramic composite armor, laminate structure, ballistic damage, residual strength, spliced ceramic plate

## Abstract

In this paper, the ballistic damage mechanism and residual bearing capacity of ceramic/backing plate armor were investigated. First, a series of lightweight armors were prepared, consisting of ceramic and ultra-high molecular weight polyethylene fiber-reinforced resin matrix composite (UHMWPE) plates, and were wrapped in a high-strength fabric. Then, the ceramic/UHMWPE armors were hit by one or two bullets, and finally subjected to compression testing. The results showed that the main failure mode of integral ceramic/UHMWPE armors was ceramic brittle fracture. Many zigzag patterns on the compression curve indicated that the specimens had undergone the stages of crack propagation, ceramic fragment reorganization, plastic deformation of UHMWPE backing plate, interlaminar tearing, and overall fracture. The failure of spliced ceramic/UHMWPE armors was mainly due to the dislocation between ceramic sheets; the smooth compression curves indicated that there was no recombination of ceramic fragments and obvious interlayer debonding during the compression. Under the maximum load, each ceramic/UHMWPE armor with ballistic damage did not suddenly break and fail. The structure and thickness of ceramic plates all had an impact on residual strength: under the same structure, the greater the thickness, the greater the residual strength, but the relationship between them was not linear; under the same thickness, the residual strength of the spliced ceramic/UHMWPE armor was higher. The residual strength was also related to the number of shots: after two bullets hit, its value was only one-third of that after one bullet hit.

## 1. Introduction

Ceramic materials have the characteristics of high hardness, low density, wear resistance, but their ability to withstand tension and bending is small. Therefore, when a ceramic material is used to resist bullet penetration, it needs to be equipped with a tough back plate [1]. Due to its excellent properties such as high temperature resistance and impact resistance, it has especially attracted attention in the defense of armor-piercing firebomb and projectile fragments [2]. In general, ceramic/backing plate armors consist of a hard brittle ceramic facing and deformable backing materials. The difference in physical properties of each layer interferes with the penetration of the projectile (jet). The ceramic facing the projectile destroys the projectile tip, slows it down and transfers the load to a large area of the backing; the backing supports the ceramic so that the broken ceramics cannot be scattered and sprayed to cause secondary damage. At the same time, the projectile is brought to rest through the good self plastic deformation of the backing [3,4,5]. In the fields of armed helicopters, military armored vehicles, and body protective equipment, people have been looking for the most effective way to achieve the optimal armor configuration and the lowest area density as possible, and trying to use lightweight and low-cost ceramic/backing plate armor to resist high-end threats [6,7]. The research on material characteristics, armor structure, and ballistic damage mechanisms has been carried out continuously. Light composite armor is constantly being designed and prepared, and the relationship between the residual penetration depth and the thickness of the ceramic panel has also been revealed [8,9].

The ceramic/backing plate armor is similar to a laminate structure and contains two or more different materials. Compared with metals, the damage propagation law of laminates is difficult to predict. Under impact load, the deformation of the laminate is the reason why it can absorb energy. The laminates preferentially exhibit plastic deformation owing to the absorbance of impact energy [10,11,12]. According to the results of shooting experiments, the damage forms found in some ceramic/backing plate armors include: visible plate burst damage, backing plate concave (convex), large deformation damage, interlayer delamination damage, etc. There have also been invisible plastic deformation, micro cracks, and other damages [13,14].

Residual strength refers to the maximum bearing capacity of a cracked structure, which generally depends on the material properties, initial crack length, and service time. The purpose of residual strength measurement is to predict whether the structure can withstand damage tolerance loads under certain damage conditions without catastrophic failure [15,16,17]. Residual strength research belongs to the strength research of damaged and defective components and is an important consideration in safety assessment. Through the test and analysis of the actual bearing capacity of damaged and defective structures, a reference can be provided for inferring the service life of the structure. At present, whether in the military or other areas with protection needs, the requirements for protective armor that can ensure the safety of personnel and equipment are becoming higher and higher. For example, a helicopter hit by bombs during a mission needs to return safely, and an armored vehicle attacked by an explosive must still maintain the ability to survive and fight on the battlefield. The residual strength can be used to evaluate the protective ability of composite armor after a bullet attack, providing an effective research method for equipment protection and personnel survival.

In this study, the ceramic/backing plate armor samples were prepared by using ceramics as the front plate and ultra-high molecular weight polyethylene (UHMWPE) fiber-reinforced resin matrix composite as the backing plate, here, named as ceramic/UHMWPE armor. There were two types of ceramic plate: integral or spliced. Then, they were arranged for shooting experiments, and a subsequent compressive test; the relative experimental phenomena were analyzed, and the failure mode and crack propagation law were discussed and compared for the different ceramic/UHMWPE armors Finally, residual strength was introduced to access the safety of the ceramic/UHMWPE armors with impact damage, and the affecting factors also were discussed. The present study will provide a methodology and some reference data for obtaining the optimum design of ceramic/UHMWPE armor with high safety performance.

## 2. Experiment

### 2.1. Experimental Materials and Sample Preparation

Carbon fiber-reinforced silicon carbide (C/C-SiC) ceramics were prepared by the precursor conversion method, which was used as the front plate of ceramic/UHMWPE armors. First the mixture of chopped carbon fiber and other materials was molded at room temperature, and carbonized at high temperature to obtain low-density carbon/carbon (C/C) composites, that is, chopped carbon fiber preforms. Then, they, as the skeleton material of C/C-SiC ceramics, were impregnated and cured in the precursor of Polycarbosilane solution (PCS). Finally, the solidified green bodies were treated at high temperature. The process was given in the literature [18,19]. After 11 cycles of “impregnation-solidification-pyrolysis”, the density of C/C-SiC ceramics was about 1.98 g/cm^3^ [20].

The C/C-SiC ceramic samples of two kinds of structure were prepared by using the self-made mold. One was the large-sized ceramic sheets (side length more than 200 mm); the other was the small-sized regular hexagonal ceramic sheets (side length 10~100 mm). They all had three different thickness: 8 mm, 10 mm, and 12 mm. In the ceramic/UHMWPE armor, one large-sized ceramic sheet acted directly as an integral ceramic plate, and some small hexagonal ceramic sheets needed to be combined to form a spliced ceramic plate, each splicing surface being closely matched.

The backing plate was made of UHMWPE fiber-reinforced resin matrix composites with high elongation, which was provided by Xianning Haiwei composite products Co., Ltd. The UHMWPE plate was prepared by paving uni-directional (UD) cloths in the 0°/90° direction, impregnating with resin, and hot pressing. The number of layers of UD cloth was decided according to the required thickness. The test density of the UHMWPE plate was 0.97 g/cm^3^.

The ceramic/UHMWPE armor was prepared by hot pressing. It was composed of a ceramic plate and a UHMWPE backing plate, and silicone rubber was the adhesive [20]. A high-strength fabric was adhered around and on the surface of the laminated armor body as an anti-cracking layer. The area density of ceramic/UHMWPE armor was controlled to not exceed 4.4 g/cm^2^ to ensure it was lightweight. According to the early research, when the thickness ratio of front plate to backing plate was about 1~2, the laminated armor had better anti-penetration performance [1,21]. On the premise of ensuring the preparation of equal area density ceramic/UHMWPE armor, the appropriate thicknesses of several ceramic plates and backing plates were designed. The sample sizes are given in Table 1. The anti-cracking cloth not only enhanced the protection effect but also prevented ceramic fragments from splashing and injuring people during shooting and the compression test [22].

### 2.2. Targeting Experiment

The position spacing of the test device in the target impact test system was arranged according to the American MIL-PRF-46103E (Armor: lightweight composite/level III-Class 2B) standard [23]. The distance between the shooting platform and the test target was 30 m. The ballistic gun was used, and the bullet was 12.7 mm armor-piercing projectile. The incident velocity was 488 m/s. The bullet velocity was measured by the speed measuring target. Each ceramic/UHMWPE armor as test target was fired by one or two bullets.

The integral ceramic plates were controlled not to be too large to ensure the uniform quality and performance. Thus, the size of the integral ceramic/UHMWPE armor samples was also smaller (210 mm × 210 mm). In order to avoid size effect, one sample was only hit by one bullet. The spliced ceramic plates were composed of many small hexagonal ceramic sheets; the splicing size was not limited, so that the prepared ceramic/UHMWPE armor samples were also larger in size (400 mm × 250 mm), and one sample could be tested with one bullet or two bullets.

### 2.3. Compression Test of Residual Strength

The loading equipment was MTS/Sans SHT600 microcomputer controlled electro-hydraulic servo universal testing machine (MTS Systems Corporation, Shanghai, China) [24,25]. The test bench was equipped with spherical bearing pad, which made the center automatically adjustable to ensure the axial loading of the sample. By using the residual strength evaluation test method of composite laminates after impact in ASTM D7137/D7137M-2017 standard [26], the compression failure test was performed on the ceramic/UHMWPE armor samples with ballistic damage. According to the test specifications and requirements of ASTM standard, the anti-instability support fixture was designed to ensure the reliability of the experimental effect. The impacted sample was installed on the plane of the test bench through the fixture, and the load was slowly and evenly applied until the specimen was completely destroyed.

During the test, the computer collected the test data. The data transmission speed was 12 MB/s. The data of loading time, force, and displacement were recorded, and the relationship curve between compression load and deformation displacement was drawn. The whole process from the beginning of loading to the destruction of the sample was recorded for further observation and analysis of experimental phenomena.

## 3. Results and Discussion

### 3.1. Experimental Analysis of Shooting Test and Ballistic Damage Mechanism

The front and back shapes of the ceramic/UHMWPE armors after the shooting experiment are shown in Figure 1. There is no large area of ceramic peeling off at the impact points; the layers are well bonded without obvious interlayer debonding phenomenon; the overall shape of the specimen remains unchanged, and the frontal impact point corresponds to the back convex position; the targets were not penetrated by the bullets. The heights of the back convex are shown in Table 1. The results show that the ballistic performance of the ceramic composite armors meets the requirements of the MIL-PRF-46103E ballistic standard [23].

The bullet marks on the front surface of the armors in Figure 1a,b are compared. It is found that there is obvious ceramic debris at the impact point of the integral ceramic plate, and the anti-cracking cloth around the impact point is a little convex. Thus, it can be inferred that the ballistic damage mode of the integral ceramic plates after a bullet impact is mainly ceramic crushing and crack propagation. However, the broken ceramic fragments were separated from the integral plate and stuck together with the anti-cracking cloth, which resulted in some uneven zones and bulges on the surface of the anti-cracking cloth Conversely, there is no obvious ceramic debris at the impact point of the spliced ceramic plate, the anti-cracking cloth around the impact point is flat, and there is only some less obvious indentation on the surface. It is speculated that the damage of the spliced ceramic plate after a bullet impact is mainly due to the dislocation between the small ceramic sheets; the impact force on the impact point spreads to the adjacent ceramic sheets, and the change of their relative position consumes a lot of energy, which reduces the tendency of the ceramic sheets to break. The indentation on the anti-cracking cloth is caused by the change in the relative position between the ceramic sheets.

The shape and size of the back convex lumps in Figure 1a,b are compared. It is found that the area and height of the convex on the back surface of the integral ceramic target are small, while those on the back surface of the spliced ceramic target are relatively large. This can be explained by the damage mechanism of the ceramic plate. The damage of the integral ceramic plate is mainly fragmentation, most of the impact energy is propagated along the radial, circumferential, and thickness directions in the ceramic plate, and a lesser part is transferred to the backing plate, so the back convex area and height are smaller. The spliced ceramic plate absorbs the impact energy mainly through the dislocation between the ceramic sheets when the target is impacted. However, the number of dislocations is limited, and the energy absorbed by the dislocation is also limited. Part of the energy is easily transferred to the backing plate, so there is a big bulge on its back surface.

### 3.2. Experimental Analysis of Compression Test and Failure Mechanism

Considering the diversity of the damage modes of composite armor materials, it is difficult to select simple characterization parameters such as crack length to determine the damage degree. Therefore, the residual strength evaluation test method of composite laminates after impact in ASTM D7137/D7137M-2017 standard was used to conduct compression failure tests on the composite armor specimens with impact damage after target shooting [25]. Their damage propagation processes, deformation modes, and final failure modes during compression failure were analyzed and compared. Finally, the residual strengths were calculated [26].

Figure 2 shows the designed anti-instability support fixture and its application in the compression test. The spacing between the components in the support fixture can be adjusted according to the size of the test sample, and a set of vertical guide plates are added to restrain the tested sample, so as to avoid instability or bending of the sample during loading.

The compression test was carried out at room temperature. The samples were slowly loaded and the load/displacement curves were observed on the computer screen. During the whole loading process, the fixture was not damaged, such as loosening and tilting, and the ceramic composite target samples also did not undergo deflection, distortion, or other unstable phenomena, indicating that the test process was reliable and the test data were valid.

Because the anti-cracking layer was wrapped outside the target plate, the crack propagation in the target plate during the compression failure process could not be seen immediately. The failure process was only judged by carefully listening to the sound of crack propagation. The actual situation of crack propagation could be observed only after the experiment was completed and the anti-cracking layer was removed.

#### 3.2.1. Composite Armors with Integral Ceramic Plate

(1) Experimental phenomena and analysis in compression testing

Figure 3 shows some real-time images taken during the compression failure of the ceramic/UHMWPE armor with integral ceramic plate. As shown in Figure 3, within 1 min after loading, the morphology around the bullet hole did not change significantly, but the cracking sound of the ceramic plate was heard, presumed to be the friction sound of residual fragments in the trajectory after bullet impact.

At 3.5 min, some tearing marks were seen around the bullet hole, and the bullet hole also appeared laterally deformed; the crackling sound changed from intermittent to continuous, and the sound became clearer and clearer. It was judged that at this stage, the crack began to slowly expand outward along the circumference of the bullet hole and the overall deformation of the sample was not obvious, but the anti-cracking layer around the bullet hole appeared bulging, which was speculated to be the crack propagation of the damaged ceramic fragments around the bullet hole.

After 4.5 min of loading, the bulging area and range on the surface expanded rapidly, showing a clear trend of lateral failure. During the test, small ceramic fragments broke out continuously from the bullet hole, the fracture sound was strengthened and sustained, and the brittle and short ceramic cracking sound was mixed with the continuous interlayer tearing sound. Some continuous lateral bulges were formed on the anti-cracking layer of the sample surface and extended to both side edges of the specimen, indicating that the damage had begun to expand laterally in the specimen.

After 6 min of loading, the area, range, and direction of the bulges no longer changed, and it was judged that the sample had been completely damaged.

(2) Failure morphology analysis

The specimen after the compression failure test and its failure morphology after removing the anti-cracking cloth is shown in Figure 4. Figure 4a is the front of the specimen, that is, the surface of the ceramic plate impacted by the bullet. It can be seen that even if the compression load is removed, the bumps and folds formed still exist on the surface. This proves that the crack propagation led to the brittle fracture of the ceramic plate, and the bumps and folds are the manifestation of the separation of the ceramic fragments bonded to the anti-cracking cloth from the backing plate. The compression wrinkles on the surface anti-cracking layer were greatly reduced because the natural gap between the ceramic fragments was restored after the load was removed and the anti-cracking cloth layer also was roughly flat, which further confirmed that there was no debonding between the anti-cracking cloth and the ceramic plate.

Figure 4b shows the back of the specimen after unloading, that is, the back surface of the UHMWPE plate. It can be seen that the backing plate was bent along the transverse center line of the bullet hole, and the surface anti-cracking cloth was stacked laterally at the bend. This proves that the UHMWPE plate had undergone the large plastic deformation under normal pressure until it broke. The deformation and fracture caused the anti-cracking cloth to be completely debonded from the backing plate, and the layered stacks on the back surface could not be recovered even after unloading. The anti-cracking cloth in the vertical direction did not wrinkle, indicating that the shear pressure was not enough to cause obvious plastic deformation of the back plate.

After the anti-cracking cloth layer wrapped around the armor sample was peeled off, the final damage of the ceramic plate and the sandwich structure can be seen as shown in Figure 4c,d. When the anti-cracking cloth was removed, some ceramic fragments stuck to it and were taken away. Therefore, the left of Figure 4c shows the fractured shape of the inner layer of the ceramic plate. After the ceramic fragments were carefully separated from the anti-cracking cloth and replaced on the front surface of the ceramic plate, the damage morphology is restored and is shown in the right of Figure 4c.

As can be seen from Figure 4c, the parts near the impact point have been broken into fragments. In the right of the figure, within 100 mm around the center of the bullet hole, there are only a few radial cracks seen on the surface of the ceramic plate. In the left of the figure, the inner layer of the ceramic plate, the cracks along the circumference of the bullet hole can be classified into radial cracks, circumferential cracks, and crack cones extending to the back; the radial cracks and the circumferential cracks intersect to form a flower-like ceramic cone section; the radial crack propagation is mainly concentrated in the inner ring of ceramic cone, the circumferential crack propagation is mostly in the outer ring, and the crack distribution is roughly uniform. If the restraining effect of the anti-cracking cloth is ignored, the main causes for ballistic impact damage of the ceramic plate can be summarized as follows. At the initial stage of bullet impact, the velocity was high and the time was short, so that the area around the impact center was not sensitive to impact damage and there were few cracks (if there are, there are only a few radial cracks) on the surface of the ceramic plate because it was too late to make a failure response. As the bullet further penetrated the ceramic plate, the motion resistance increased and the bullet speed slowed down rapidly, so that the contact time with the trajectory was longer than that in the initial stage and the stress wave generated from the trajectory had enough time to propagate through the width and thickness of the target plate; therefore, a crack cone was formed. When the diameter exceeded 100 mm diameter, the crack propagation was directional, mainly extending along the horizontal direction and about ±45° direction (still intersecting in the horizontal direction); the crack propagation surface extended from the center of the bullet hole to the edge of the specimen. These transverse cracks perpendicular to the loading direction eventually led to the overall failure of the ceramic/UHMWPE armor.

Figure 4d shows the side of the ceramic/UHMWPE armor without the anti-cracking cloth, and the debonding and delamination between the two plates can be seen; a thin blade was inserted to detect the interlayer separation. It was found that the plates had been completely debonded after the compression failure test, but the largest interlayer gap was not in the fracture site of the sample but near the fracture site. This shows that in the compression test, due to the different plastic deformation degree in the backing plate and different crushing degree in the ceramic plate, the debonding on the non-fracture zone first started and became more and more serious, while on the fracture zone, the fracture first occurred and was followed by debonding, the debonding parts at the fracture being very shallow.

(3) Experimental data analysis

The loading force and displacement data recorded in the compression experiment were processed. The amount of collected data was too large, so it was adopted that to reduce the amount of data, 10 consecutive data were divided into a group and the median was taken. In this way, the reliability of the data was also improved. Based on the processed data, the load force/displacement curve was drawn, as shown in Figure 5.

There are three test curves in Figure 5. They respectively correspond to the composite armor samples listed in Table 1, which are composed of the integral ceramic plate and UHMWPE backing plate of different thicknesses. Before the compression test, all samples were impacted by a bullet at high speed. The shapes of the force/displacement curves are basically similar in Figure 5, although the thickness of the plates constituting each sample is slightly different, which indicates that the compression failure mechanism is the same.

The loading process is the most noteworthy stage, which reflects the remaining bearing capacity and failure mode of the ceramic/UHMWPE armors with ballistic damage. It can be seen from Figure 5 that during the loading stage, the overall curve changes smoothly, the slope first gradually increases and then basically remains unchanged, and the curve does not rise until the maximum compressive load. At the beginning of loading, the displacement of the sample was very small; this was at the stage of eliminating the gap, which resulted from the insufficient initial contact between the test sample and the fixture in the installation process. Under the action of the load, the various parts gradually contacted, the gaps disappeared, and the slight deformations began. When the load exceeds 10 kN, the force/displacement curve changes to a zigzag shape, which is similar to the yield phenomenon of plastic materials. In fact, it was the process of gap adjustment and the rearrangement of ceramic fragments under the action of tangential force, which was in harmony with the deformation of the UHMWPE backing plate. After that, the force/displacement curve shows a large slope without obvious fluctuations, which proves that the rearranged ceramic fragments once again played the role of the internal ceramic plate, and the deformation resistance of the whole armor improved. Before the maximum load is reached, the curve fluctuates to a zigzag shape again, indicating that the rearranged ceramic fragments began to produce new cracks and gradually separate from the backing plate, and that the backing plate also began large plastic deformation until the fracture. Briefly, the armor sample underwent the stages of ceramic fragment reorganization, plastic deformation, interlayer debonding, crack propagation, and fracture caused by compressive loads.

The top of each force/displacement curve in Figure 5 shows some repeated zigzag fluctuations. This shows that after the maximum load was reached, the broken ceramic fragments were repeatedly rearranged to form a pseudo-integrity and then broken again along with interlayer crack and backing plate deformation, and the deformation resistances of the whole armor was in a state of high and low fluctuations. After such a period of time, the force/displacement curve decreases rapidly, indicating that the specimen had completely failed.

From the beginning of the test, the “squeaking” sound of ceramic fragmentation was heard from the sample, which was due to the relative movement and friction between the ceramic debris remaining in the trajectory caused by bullet penetration. The ceramic debris belongs to the failed part.

With the increase in load, the repeated cracking sound, composed of the brittle “squeaking” and “popping”, continued to come out. This showed that new fragmentation continued to appear in the ceramic plate during the crack propagation process. In the test, the sound of interlayer tearing was also constantly heard, especially when the maximum load was approaching, the sounds of ceramic cracking and interlayer tearing were intertwined together.

The change in sound is consistent with the fluctuation in the force/displacement curve. The fragmentation of the ceramic plate absorbed a certain amount of energy and the reorganized fragments increased the deformation resistance of the sample, which was an important reason for the fluctuation of the loading force. In the fracture stage of Figure 5, the maximum load force fluctuated many times continuously, and the sound of interlayer tearing was heard constantly at this stage of the test, indicating that the ceramic plate had been severely broken and the backing plate had been greatly deformed, the whole target gradually lost its bearing capacity. It is because of the large deformation of the backing plate and the interlayer debonding, which consume a part of the load work, that the sample has the ability to withstand such a load. It can be seen from the force/displacement curve in Figure 5 that all samples had a deformation of about 2.5~3 mm under the maximum load. Therefore, the fracture failure of ceramic/UHMWPE armors after ballistic damage is not sudden. The damaged ceramic composite armors can still work safely through deformation under a certain load.

#### 3.2.2. Composite Armors with Spliced Ceramic Plate

(1) Experimental phenomena and analysis in compression testing

The morphology change in the ceramic/UHMWPE armor with the spliced ceramic plate under compressive load is shown in Figure 6. When the compression load reached 50% of the final failure load, the intermittent crack propagation sound from the ceramic/UHMWPE armor began to be heard, but the morphology around the bullet hole in Figure 6b seemed to show no significant change in comparison with the original state in Figure 6a. With the increase in and continuity of fracture sound, the crack in the area around the bullet hole began to slowly expand in the lateral direction. When the load reached more than 80% of the final failure load, the obvious bulges on the anti-cracking cloth were seen from the sample in Figure 6c, and the small ceramic particles also popped out of the bullet hole during the test. As the load continued to increase, the sound of ceramic chipping and interlayer tearing began to be heard in the sample, indicating that the damage had begun to expand inside the specimen. As shown in Figure 6d, a continuous horizontal bulge formed on the anti-cracking cloth layer, extending to one side of the sample. The trajectory was no longer a regular circle, and there were debris bulges with different heights inside, making the surface of the trajectory very irregular. After the load reached the maximum value for a period of time, the specimen failed.

(2) Failure morphology analysis

Figure 7 shows the morphology of the sample after the compression failure test was performed and the anti-cracking cloth was removed. Before the compression test, the spliced ceramic/UHMWPE armor sample withstood the impact of two bullets, and two bullet holes were left on the upper and lower parts of the target plate, respectively.

Figure 7a is the front face of the sample, that is, the spliced ceramic plate. After unloading, the front surface was generally flat, the bulge formed around the bullet hole still existed, and the compression wrinkles on the surface of the anti-cracking cloth had not healed, indicating that the ceramic plate was not a completely brittle fracture and the failure was caused by the dislocation of the ceramic sheets along the joints. Even when the load was removed, the ceramic sheets still no longer returned to their original position because they had already been in a state of dislocation equilibrium. The creases on the anti-cracking cloth were caused by the protrusions on the edge of the dislocation, and there was no debonding between the anti-cracking cloth and ceramic sheets.

Figure 7b shows the back of the sample, a UHMWPE backing plate. It can be seen that the backing plate was bent along the transverse center line of the specimen (perpendicular to the direction of compressive load), rather than along the center line of the bullet hole. However, the surface anti-cracking cloth was stacked horizontally along the center lines between the two bullet holes, indicating that the UHMWPE back plate preferentially had large bending deformation between the two bullet holes, which caused the anti-cracking cloth to appear as unrecoverable local debonding even after unloading. There was no wrinkle of the anti-cracking cloth in the vertical direction, indicating that there was no sufficient shear force to cause shearing deformation and debonding of the backing plate during bending deformation.

After the outer anti-cracking cloth was torn off, the final damage morphology of the spliced ceramic plate on the front and the sandwich structure on the side of the specimen could be seen, as shown in Figure 7c,d. It took a lot of force to remove the anti-cracking cloth from the ceramic sheet because they adhered very tightly without debonding. It can be seen in the left part of Figure 7c that a large amount of adhesive remains on the ceramic sheets. The crack on the residual adhesive indirectly reflects the damage of the spliced ceramic plate. The crack propagated radially within 100 mm of the impact center point, which was consistent with that of the integral ceramic plate. The compression failure cracks extended laterally from the center of the two impact points to both sides of the specimen because of the upper and lower compression limiting the crack propagation direction. In addition, there was a through crack between the two impact points, indicating that the impact point is the source of crack propagation and the main reason for reducing the remaining bearing capacity of the target plate. There were no visible cracks in the other parts of the adhesive surface, but some block imprints could be found, which reflected the position change of the spliced ceramic sheets.

The right part of Figure 7c shows the ceramic surface after part of the adhesive has been removed. The removed adhesive carried away some surface ceramic fragments. Some radial and circumferential cracks were seen in the spliced ceramic plate near the impact point, which were the result of the stress waves generated by the penetration movement of the warhead. The ballistic damage mechanism is the same as that of the integral ceramic plate. However, the number and density of cracks caused by ballistic damage were much less than those in the integral plate. This is mainly because the shear stress generated by the penetration of the bullet made the ceramic sheets in the spliced ceramic plate easier to move relative to each other. The relative motion absorbed a large amount of impact energy, so that the ceramic sheets were no longer broken, and only changed the position.

Figure 7d shows the side of the composite armor plate with the anti-cracking cloth removed. In order to detect the debonding and delamination between the layers, a thin blade was inserted. It can be seen that the blade was not completely inserted into the interlayer at the side, indicating that there was no interlayer debonding between the ceramic plate and the backing plate after the compression failure test. The reason should be that the size of the ceramic sheets used for splicing was small and they were dislocated and misaligned with the deformation of the backing plate under stress. The backing plate was seriously deformed and bent in an arch shape. In the test, the tearing and breaking sound of the backing plate was heard,, but no surface cracks are seen in Figure 7d owing to the adhesive adhered on the surface and covered the fracture cracks.

(3) Experimental data analysis

The size of the spliced ceramic plate was large, which made it possible to study the residual strength of the ceramic/UHMWPE armor after multiple strikes. The processing method of data from the compression experiment is the same as above. The load force/displacement curves are shown in Figure 8. They represent the compression test of the spliced ceramic/UHMWPE armor that suffered one or two bullets, respectively.

The force/displacement curves in Figure 7 are generally smooth, showing a 
gradual increase in the slope until the maximal load is reached. Compared with Figure 4, there are fewer zigzag fluctuations 
at any stage in the force/displacement curve, which is similar to the 
compression test curve of the overall material. Such a curve shape is also 
consistent with the failure morphology presented in Figure 6. There are relatively few fragments on 
the spliced ceramic plate after bullet impact, and there is almost no 
interlayer debonding. For a long period under the initial load, the slope of 
the curve is stable at a small value. It can be considered that the dislocation 
of the ceramic sheets caused by bullet impact did not affect the composite 
target plate as a whole to bear the compressive load. It can be seen from Figure 8 that both curves have a phase with an 
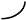
-shaped turning slope. From this stage, a very crisp and bright cracking sound “bang” 
was heard intermittently during the test. With the increase in load, that is, 
the rapid rising stage of the curve, the sounds of cracking, crushing, and 
friction were continuous, which indicated that the crack was expanding inside 
some ceramic sheets, while the other ceramic sheets were misaligned in the 
tangential direction. When the load reached the maximum value, the fracture 
occurred quickly. This is because the fracture zone caused by the ballistic 
damage and crack propagation in the ceramic sheets had no ability to bear any 
load. The load must propagate along the unconstrained lateral direction between 
ceramic sheets, which is the reason for the formation of the final fracture 
zone.

In Figure 8, the thick red solid line is the test curve of the sample fired by two bullets. There is a small zigzag wave in the rising phase of the curve, which just explains that the crack propagation caused by the two impact trajectories was not synchronous. The cracks around one trajectory first expanded and the ceramic sheets failed to load, and all the loads were quickly transferred to the other parts so that the compression damage under the influence of two trajectories merged into one at this time. It can be seen from the thick curve in Figure 8 that the slope of the rising phase after the zigzag wave is slightly reduced compared with the slope of the previous curve, indicating that the deformation resistance of the ceramic/UHMWPE armor that has been partially compressed and damaged began to decrease. When the maximum load is reached, there is a section close to a straight line at the top of the compression curve. This shows that the ceramic/UHMWPE armor after multiple blows (multiple damages) had a process of stress dispersion, transmission, and mutual offset in the state of compression failure, so that the fracture would last for a period of time and avoid sudden failure.

The force/displacement curves in Figure 8 also prove that the compression failure mode and residual bearing capacity of the spliced ceramic/UHMWPE armor is different from those of the integral ceramic/UHMWPE armor, just as their ballistic damage forms are different. The ballistic impact damage of the integral ceramic/UHMWPE armor is mainly the ceramic crushing cone that resulted from the stress wave, and that of the spliced ceramic/UHMWPE armor is mainly the dislocation between the ceramic sheets. In the process of compression failure, the spliced ceramic/UHMWPE armor had no ceramic fragment reorganization, no obvious yield stage, and no interlayer debonding, while the integral ceramic/UHMWPE armor exerted its bearing capacity as a whole through self reorganization. In the same way as the integral ceramic/UHMWPE armor, the fracture of the spliced ceramic/UHMWPE armor did not happen suddenly. Therefore, the two kinds of ceramic/UHMWPE armor can maintain their working state through deformation under compression load, so as to buy time for the protection of life and property.

#### 3.2.3. Calculation of Residual Strength and Influencing Factors

After the compression test of the ceramic/UHMWPE armors that had suffered bullet impact, the residual compressive strength of each sample can be calculated. The calculation formula is shown in Equation (1).
(1)RCAI=PmaxA
where: *R*_CAI_ is the residual compressive strength (MPa); *P*_max_ is the maximum compressive force borne by the sample before failure (N); *A* is the cross-sectional area of the sample (mm^2^). The calculation results are shown in Table 1. The relationship between compressive strength and ceramic plate thickness is shown in Figure 9.

By comparing the data in Table 1, it is known that the greater the plate thickness is, the greater the residual strength is, whether the ceramic/UHMWPE armor is with an integral ceramic plate or with a spliced ceramic plate (similarly hit by one bullet or two bullets). However, when the plate thickness is increased from 10 mm to 12 mm, the residual strength does not change much, which indicates that there should be an appropriate value between the thickness of the ceramic plate and its protection ability when preparing a composite armor, and it can be selected and designed according to the protection requirements. After being hit similarly by a bullet, the residual strength of the spliced ceramic/UHMWPE armor is significantly greater, which is almost twice that of the integral ceramic/UHMWPE armor, indicating that the spliced ceramic/UHMWPE armor has higher protection ability after bullet damage. After being hit by two bullets, the residual strength of the spliced ceramic/UHMWPE armor reduced a lot, but such a small residual strength still meets the design load (>0.004782 Mpa) requirements in CCAR-25 [27].

### 3.3. Future Research Directions

The main task of this research is to experiment, analyze, and discuss the experimental phenomena, so as to reveal the projectile penetration mechanism of ceramic/UHMWPE armors and the residual bearing capacity after damage. However, the test cost is high and the number of tests is limited. In the future, it is necessary to numerically simulate the test phenomena and establish the model for optimizing the armor structure and predicting failure. The initial tests provide a basis for verifying the fidelity of the numerical model and material model parameters. The combination of the experiment and numerical simulation has been proved to be very effective in studying the failure mechanism of the armor plate [28].

In fact, when the projectile strikes the ceramic armor, part of the impact energy is converted into heat energy, which increases the local temperature of the armor material, resulting in the change of material properties. It is difficult to observe and capture these changes during the test, but this is a problem that must be paid attention to, especially when establishing the numerical model.

The flow stress model of a projectile hitting hard armor plate has been established, which is a cumulative damage fracture model considering loading history [28]. When the impact equivalent stress borne by the target plate is calculated, not only the experimental loading parameters but also the influence of thermal softening are considered, including the experimental ambient temperature *T*, room temperature *T**_r_*, and the melting point temperature *T**_m_* of the material, which are normalized to $T_{H}=\frac{T-T_{r}}{T_{m}-T_{r}}$. From the normalized temperature equation, it can be seen that the operating environment of the armor material has a certain impact on the impact stress. At extreme high and low temperatures, penetration results may show differently from those in laboratory tests. The higher the melting point of the target material, the less the impact of the working environment temperature on the penetration results, which further explains the reason for choosing high temperature resistant materials as bulletproof targets [1,2,3].

Previous studies have also proved that the impact energy absorption capacity of polyphylene, polypropylene, and their composites are affected by impact temperature, impact velocity, and strain rate [29,30]. These research results will provide guidance for the establishment of a numerical analysis model in the next step.

## 4. Conclusions

Based on the principle of being lightweight, a series of ceramic/UHMWPE armors were prepared by controlling the area density so that it did not exceed 4.4 g/cm^2^. Through the shooting test, the bulletproof performance of the designed ceramic/UHMWPE armors was proved to meet the requirements of the American bulletproof standard in MIL-PRF-46103E. Due to the different ceramic size and configurations, the ceramic/UHMWPE armors showed different properties in ballistic damage mechanism, compression failure behavior, and residual bearing capacity.

Ceramic plates with different structures showed different ballistic damage modes. The damage to the integral ceramic plate was mainly ceramic crushing due to the action of the stress wave back and forth, the petal-like ceramic cone was finally formed. The main damage form of spliced ceramic armor was the dislocation of ceramic sheets. This is because the preferential movement of adjacent ceramic sheets consumed part of the impact energy, and the residual energy was not enough to lead to the crushing of the ceramic sheets.

The configurations of ceramic plates were different, and the ceramic/UHMWPE armors after ballistic tests showed different compression test phenomenon and compression failure modes. The integral ceramic/UHMWPE armor experienced the stages of ceramic fragment reorganization, plastic deformation of the backing plate, interlayer tear, crack propagation, and fracture caused by compressive load; the zigzag phases on the compression curve proved that the ceramic fragments were reorganized to form a pseudo-integral ceramic plate, and then destroyed rapidly again. The spliced ceramic/UHMWPE armor had no reorganization of the ceramic fragments and obvious interlayer debonding. It always exerted its bearing capacity as a whole, which is proved by its smooth compression test curve with a relatively steep slope similar to that of a single uniform material.

Under the maximum load, the all ceramic/UHMWPE armors first deformed to a certain extent, and then broke, indicating that the ceramic/UHMWPE armors after bullet shooting damage could still work safely through deformation: win time for the protection of life and property.

Residual strength is an absolutely competent parameter in the structural safety assessment and life prediction. The remaining bearing capacity of the ceramic/UHMWPE armor after ballistic damage was affected by the configuration, thickness, and damage mode of the ceramic plate. After a single attack with the same working condition, the residual strength of the spliced ceramic/UHMWPE armor was greater than that of the integral ceramic/UHMWPE armor provided that their total thickness and the plate thickness of each layer was the same. Under the same working condition, after two bullets hit, the residual strength of the spliced ceramic/UHMWPE armor decreased rapidly, which was only one-third of that after one bullet hit. Similarly, after one bullet hit, the greater the thickness of the ceramic plate, whether it was integral or spliced, the greater the residual strength of the composite armor. However, as shown in the experimental data shown in Table 1, the relationship between the residual strength and the thickness of the ceramic plate does not increase linearly. When preparing protective armor equipment, the requirements of manufacturing cost, service condition, and protection level can be comprehensively considered to design an appropriate thickness of the ceramic plate.

## Figures and Tables

**Figure 1 materials-15-00901-f001:**
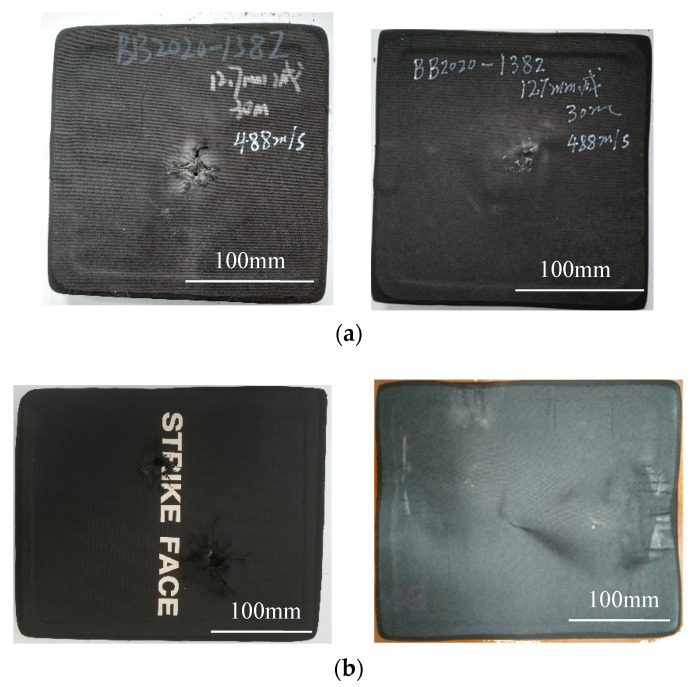
Front and back surfaces of ceramic composite targets after bullet impact. (**a**) Integral ceramic panel composite target (one bullet hit) Lift-Front, and Right-Back; (**b**) Spliced ceramic panel composite target (two bullets hit) Lift-Front, and Right-Back.

**Figure 2 materials-15-00901-f002:**
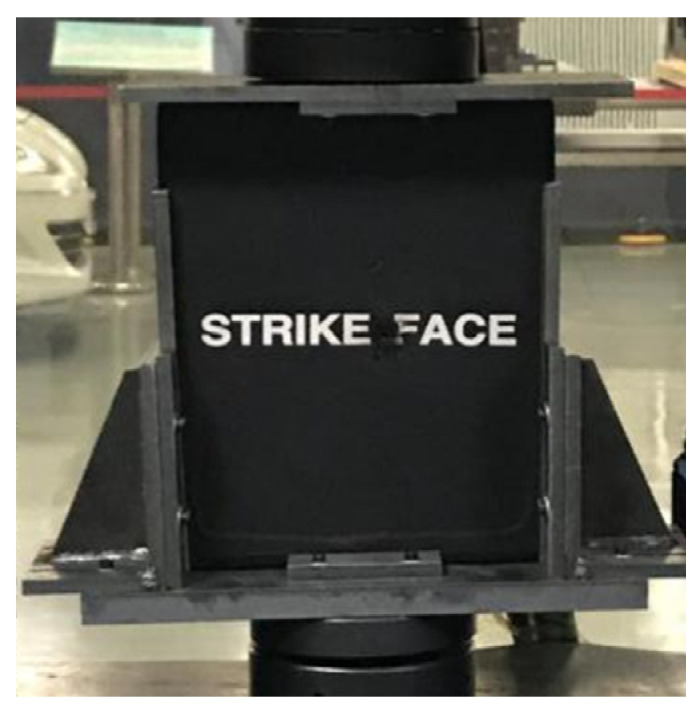
Residual strength support fixture and its application in compression experiment.

**Figure 3 materials-15-00901-f003:**
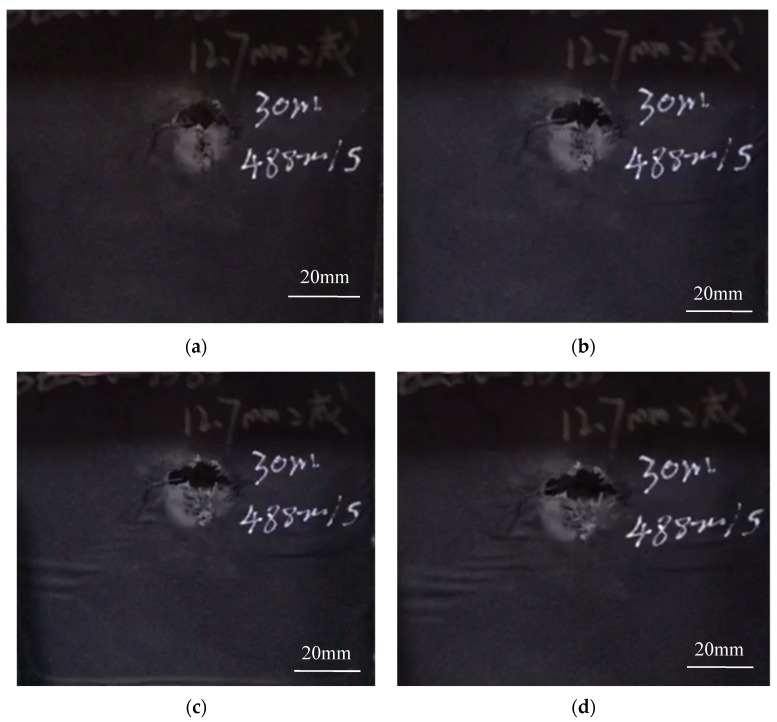
Morphology of target plate with integral ceramic panel during loading test. (**a**) at 1 min; (**b**) at 3.5 min; (**c**) at 4.5 min; (**d**) at 6 min.

**Figure 4 materials-15-00901-f004:**
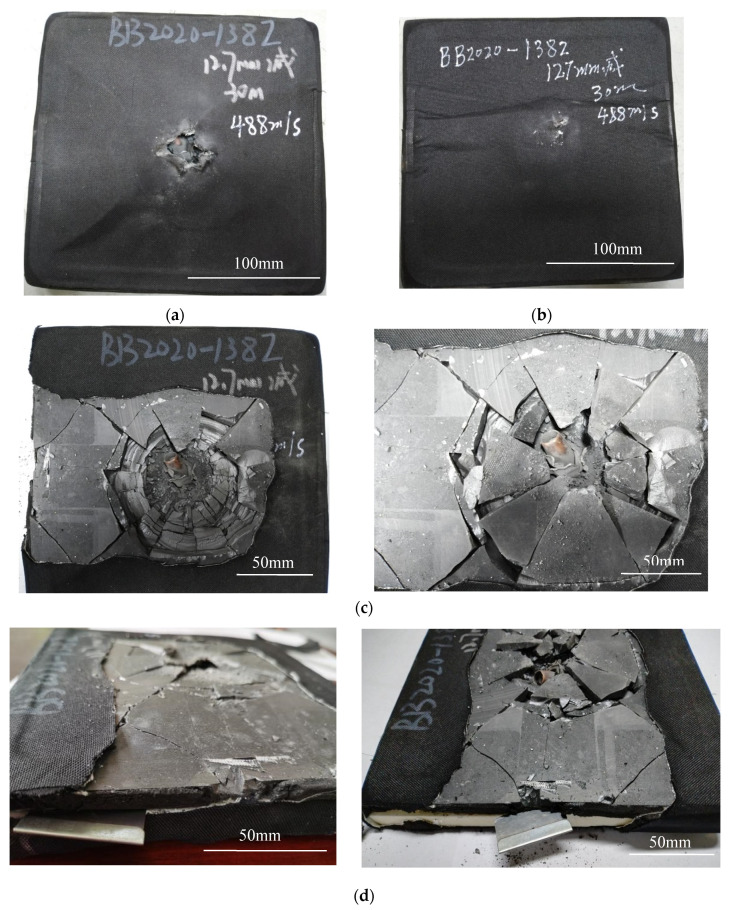
Compression damage morphology of composite target plate with integral ceramic panel. (**a**) Front face of specimen after unloading; (**b**) back face of specimen after unloading; (**c**) ceramic panel stripped of crack arrest layer: Left-some surface ceramic fragments have been taken away by the crack arrest cloth, and Right-the surface ceramic fragments have been reset; (**d**) side of composite armor plate stripped of crack arrest cloth.

**Figure 5 materials-15-00901-f005:**
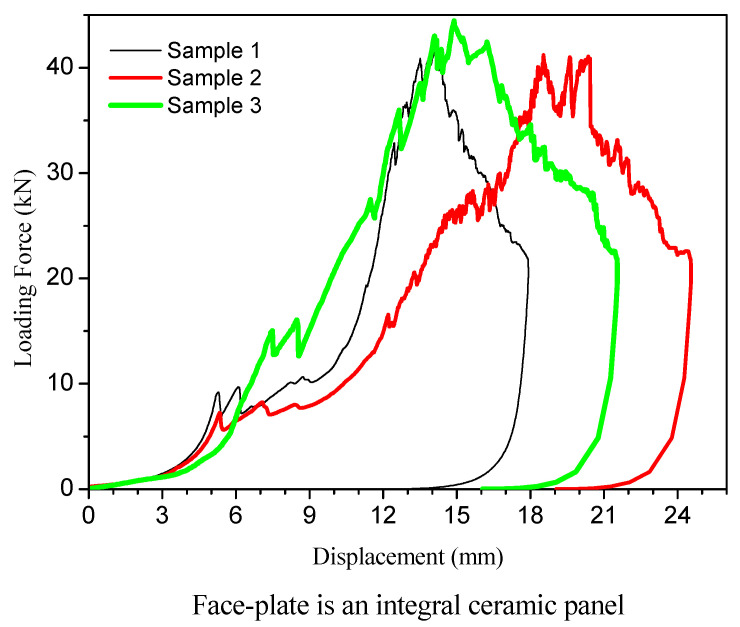
Compression test curves of samples with integral ceramic plate.

**Figure 6 materials-15-00901-f006:**
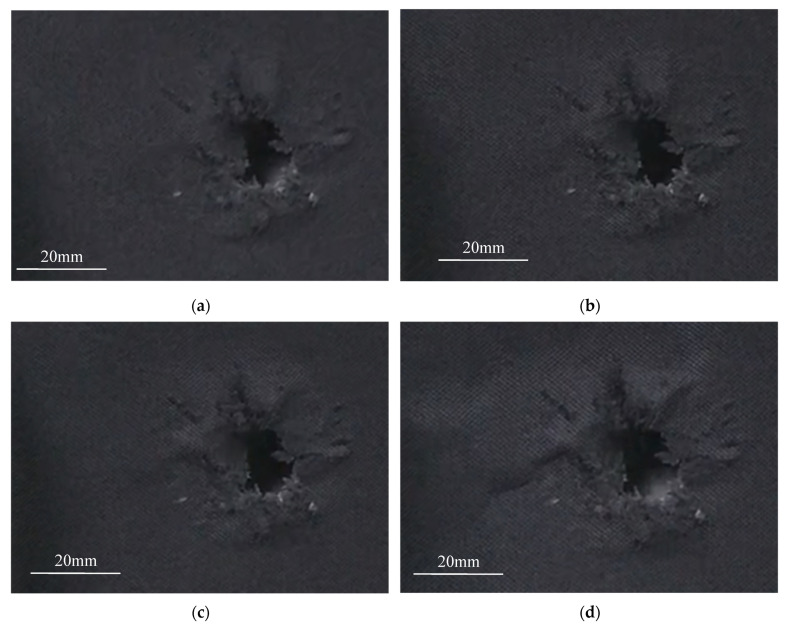
Morphology of target plate with spliced ceramic panel during loading test. (**a**) At the beginning of loading; (**b**) under 50% of the maximum load; (**c**) under 80% of the maximum load; (**d**) under the maximum load.

**Figure 7 materials-15-00901-f007:**
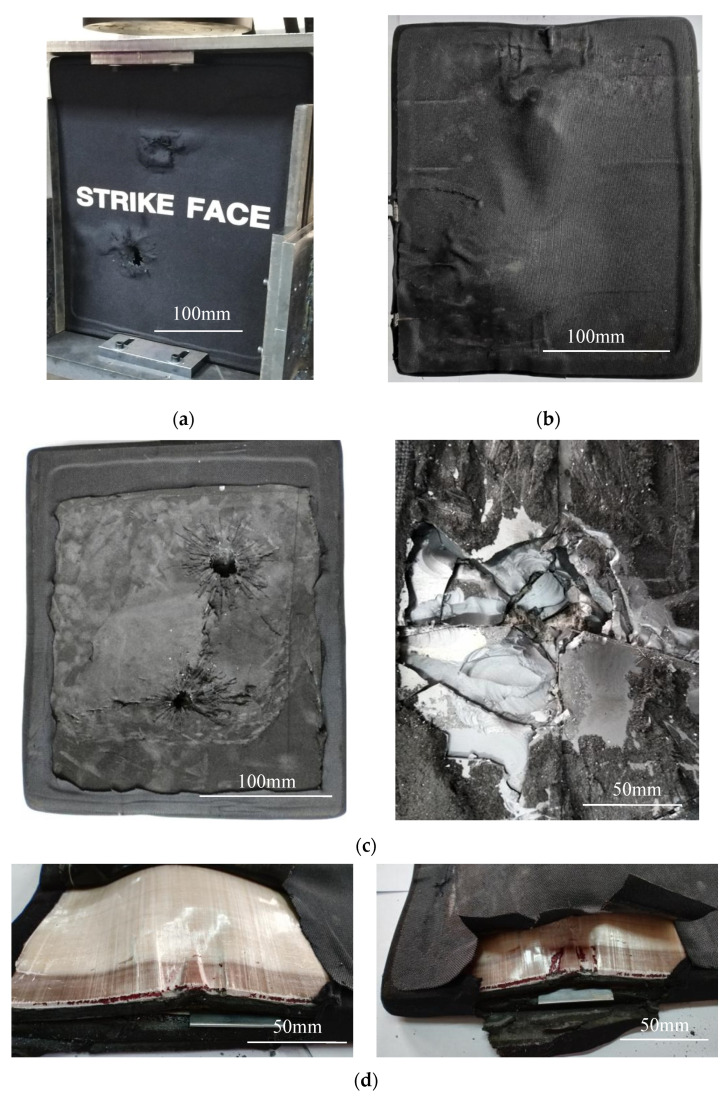
Compression damage morphology of composite target plate with spliced ceramic plate. (**a**) Front face of specimen after unloading; (**b**) back face of specimen after unloading; (**c**) ceramic panel after stripping crack arrest layer: Left-the adhesive remained on the surface, and Right-the adhesive near the impact point has been cleaned; (**d**) side of composite armor plate without crack arrest cloth.

**Figure 8 materials-15-00901-f008:**
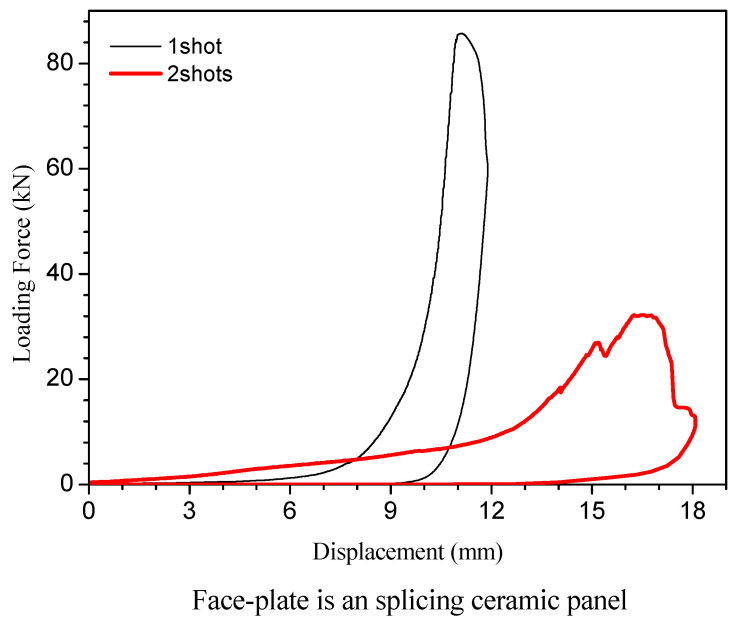
Compression load/displacement curve of composite target with spliced ceramic plate.

**Figure 9 materials-15-00901-f009:**
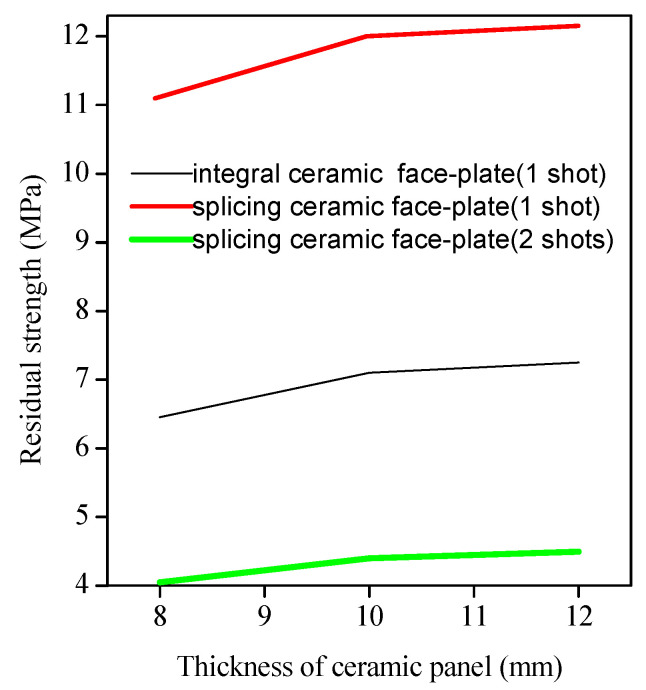
The relationship between compressive strength and ceramic panel thickness.

**Table 1 materials-15-00901-t001:** Geometric parameters and properties of composite armor samples with ceramic/UHMWPE backing plate.

Sample	Size (mm)	Ceramic Surface Plate		UHMWPE Backing Plate	Composite Armor Samples		Number of Shots	Dorsal Convex Height (mm)		Residual Strength (MPa/mm^2^)	
		Type	Thickness (mm)	Thickness (mm)	Total Thickness (mm)	Number		Measurement	Average	Measurement	Average
1	210 × 210	integral	8	$21.9_{0}^{+0.1}$	29.9	2	1	10.4	10.1	6.6	6.45
								9.8		6.3	
2			10	$18.8_{0}^{+0.1}$	28.8	2	1	9.3	9	7.0	7.1
								8.7		7.2	
3			12	$16.2_{0}^{+0.1}$	28.2	2	1	8.6	8.5	7.2	7.25
								8.4		7.3	
4	300 × 250	Spliced	8	$21.9_{0}^{+0.1}$	29.9	2	1	15.2	15.1	11.3	11.1
								15.0		10.9	
						2	2	15.7	15.5	4.0	4.05
								15.3		4.1	
5			10	$18.8_{0}^{+0.1}$	28.8	2	1	13.6	13.6	11.9	12.0
								13.6		12.1	
						2	2	13.9	14.0	4.5	4.4
								14.1		4.3	
6			12	$16.2_{0}^{+0.1}$	28.2	2	1	12.4	12.2	12.2	12.15
								12.0		12.1	
						2	2	12.8	12.7	4.4	4.5
								12.6		4.6	

## Data Availability

The data presented in this study are available on request from the corresponding author.

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
