# Peer review of "Investigation on Residual Strength and Failure Mechanism of the Ceramic/UHMWPE Armors after Ballistic Tests"

_materials, 2022, doi:10.3390/ma15030901_

Round 1
Reviewer 1 Report
Abstract - what was the goal of the study? It should be clearly stated, at least in the text.
Lines 32-36 – references needed for practically separately for each property of ceramic materials
it would be worth discussing the dependence of the projectile impact on the local temperature in the armor material and their possible influence on the change of material properties. References can be useful here:
Namık Kılıç, Said Bedir, Atıl Erdik, Bülent Ekici, Alper TaÅŸdemirci, Mustafa Güden, Ballistic behavior of high hardness perforated armor plates against 7.62mm armor piercing projectile, Materials & Design, Volume 63, 2014, Pages 427-438, https://doi.org/10.1016/j.matdes.2014.06.030.
- Alcock, N.O. Cabrera, N.-M. Barkoula, Z. Wang, T. Peijs, The effect of temperature and strain rate on the impact performance of recyclable all-polypropylene composites, Composites Part B: Engineering, Volume 39, Issue 3, 2008, Pages 537-547, https://doi.org/10.1016/j.compositesb.2007.03.003.
Also, there are no considerations on the possibility of modeling such an impact…
Line 316-317: The sample numbers in Figure 5 do not correlate very accurately with the numbers in Table 1
Line 521 : you must use subscripts at the appropriate markings
Figures 1, 3, 4, 6, 7 - it would be useful to have reference segments for a better overview of the dimensions
Lines 577-580: it is necessary to explain in more detail what the authors understand by 'a appropriate value for the thickness of the ceramic plate' and what 'the protection requirements' they took into account
What are the foreseen future directions of research
Reviewer 2 Report
Dear Authors,
Congratulations on your work, which is very interesting.
I've just a couple of concerns about your paper, thus, please see below my suggestions for improvement:
- The Introduction should be improved with more references, using the direct citation mode, i.e., describing in sum the main goals and results of some articles based on composites targetted for armour applications.
- In the Experimental/Methods, please show the setup used for each kind of tests.
- Please enlarge the figures showing the impact, because it is difficult to clearly observe the phenomena happened and described in the text.
- In Figures 5 and 8, in the Y-axis, please change KN to kN.
- Please enlarge the discussion around your results, because just two references are used to compare your results to the ones previously obtained by other Researchers in similar tests.
- The references MUST BE REVISED, because there are serious lacks of information and formatting.
Good luck.
Kind regards.
